# END-TO-END SPATIO-TEMPORAL ACTION LOCALISATION WITH VIDEO TRANSFORMERS

## ABSTRACT

The most performant spatio-temporal action localisation models use external person proposals and complex external memory banks. We propose a fully end-to-end, transformer based model that directly ingests an input video, and outputs tubelets – a sequence of bounding boxes and the action classes at each frame. Our flexible model can be trained with either sparse bounding-box supervision on individual frames, or full tubelet annotations: In both cases, it predicts coherent tubelets as the output. Moreover, our end-to-end model requires no additional pre-processing in the form of proposals, or post-processing in terms of non-maximal suppression. We perform extensive ablation experiments, and significantly advance the state-of-the-art results on four different spatio-temporal action localisation benchmarks with both sparse keyframes and full tubelet annotations.

## 1 INTRODUCTION

Spatio-temporal action localisation is an important problem with applications in advanced video search engines, robotics and security among others. It is typically formulated in one of two ways: Firstly, predicting the bounding boxes and actions performed by an actor at a single keyframe given neighbouring frames as spatio-temporal context (Gu et al., 2018; Li et al., 2020a). Or alternatively, predicting a sequence of bounding boxes and actions (*i.e.* "tubes"), for each actor at each frame in the video (Soomro et al., 2012; Jhuang et al., 2013).

The most performant models (Pan et al., 2021; Arnab et al., 2022; Wu et al., 2022; Feichtenhofer et al., 2019), particularly for the first, keyframe-based formulation of the problem, employ a two-stage pipeline inspired by the Fast-RCNN object detector (Girshick, 2015): They first run a separate person detector to obtain proposals. Features from these proposals are then aggregated and classified according to the actions of interest. These models have also been supplemented with memory banks containing long-term contextual information from other frames (Wu et al., 2019; 2022; Pan et al., 2021; Tang et al., 2020), and/or detections of other potentially relevant objects (Tang et al., 2020; Arnab et al., 2021b) to capture additional scene context, achieving state-of-the-art results.

And whilst proposal-free algorithms, which do not require external person detectors, have been developed for detecting both at the keyframe-level (Köpüklü et al., 2019; Chen et al., 2021; Sun et al., 2018) and tubelet-level (Kalogeiton et al., 2017; Zhao et al., 2022b), their performance has typically lagged behind their proposal-based counterparts. Here, we show for the first time that an end-to-end trainable spatio-temporal model outperforms a two-stage approach.

As shown in Fig. 1, we propose our **S**patio-**T**emporal **A**ction Transforme**R** (STAR) that consists of a transformer architecture, and is based on the DETR (Carion et al., 2020) detection model. Our model is "end-to-end" in that it does not require pre-processing in the form of proposals, nor post-processing in the form of non-maximal suppression (NMS) in contrast to the majority of prior work. The initial stage of the model is a vision encoder. This is followed by a decoder that processes learned latent queries, which represent each actor in the video, into output tubelets – a sequence of bounding boxes and action classes at each time step of the input video clip. Our model is versatile in that we can train it with either fully-labeled tube annotations, or with sparse keyframe annotations (when only a limited number of keyframes are labelled). In the latter case, our network still predicts tubelets, and learns to associate detections of an actor, from one frame to the next, without explicit supervision. This behaviour is facilitated by our formulation of factorised queries, decoder architecture and tubelet matching in the loss which all contain temporal inductive biases.

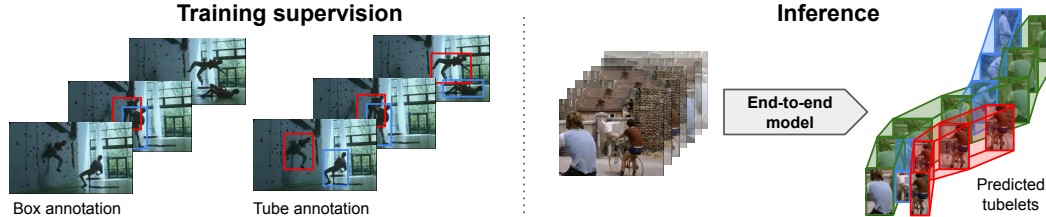

Figure 1: We propose an end-to-end Spatio-Temporal Action Recognition model named STAR. Our model is end-to-end in that it does not require any external region proposals to predict *tubelets* – sequences of bounding boxes associated with a given person in every frame and their corresponding action classes. Our model can be trained with either sparse box annotations on selected keyframes, or full tubelet supervision.

We conduct thorough ablation studies of these modelling choices, confirming the benefit of temporal inductive biases in our model design. Informed by these experiments, we achieve state-of-the-art on both keyframe-based action localisation datasets like AVA (Gu et al., 2018) and AVA-Kinetics (Li et al., 2020a), and also tubelet-based datasets like UCF101-24 (Soomro et al., 2012) and JHMDB (Jhuang et al., 2013). In particular, we achieve a Frame mAP of 45.1 on AVA-Kinetics, outperforming the best previous results achieved by a massive video foundation model (Wang et al., 2023). In addition our Video AP50 on UCF101-24 surpasses prior work (Zhao et al., 2022b) by 13.2 points. Moreover, our state-of-the-art results are achieved with a single forward-pass through the model, using only a video clip as input, and without any separate external person detectors providing proposals (Wu et al., 2022; Wang et al., 2022; 2023), complex memory banks (Wu et al., 2022; Zhao et al., 2022b; Pan et al., 2021), or additional object detectors (Tang et al., 2020; Arnab et al., 2021b), as used by the prior state-of-the-art. Furthermore, we outperform these complex, prior two-stage models whilst also having additional functionality in that our model predicts tubelets, that is, temporally consistent bounding boxes at each frame of the input video clip.

## 2 RELATED WORK

Models for spatio-temporal action localisation have typically built upon advances in object detectors for images. The most performant methods (Pan et al., 2021; Wu et al., 2022; Tang et al., 2020; Arnab et al., 2022) are based on "two-stage" detectors like Fast-RCNN (Girshick, 2015). These models use external, pre-computed person detections, and use them to ROI-pool features which are then classified into action classes. Although these models are cumbersome in that they require an additional model and backbone to first detect people, and therefore additional detection training data as well, they are currently the leading approaches on datasets such as AVA (Gu et al., 2018). Such models using external proposals are also particularly suited to datasets such as AVA as each person is exhaustively labelled as performing an action, and therefore there are fewer false-positives from using action-agnostic person detections compared to datasets such as UCF101 (Soomro et al., 2012).

The accuracy of these two-stage models has further been improved by incorporating more contextual information using feature banks extracted from additional frames in the video (Wu et al., 2022; Pan et al., 2021; Tang et al., 2020; Wu et al., 2019) or by using detections of additional objects in the scene (Arnab et al., 2021b; Baradel et al., 2018; Wang & Gupta, 2018). Both of these cases entail significant extra computation and complexity to train additional auxiliary models, and to precompute features from them that are then used during training and inference of the localisation model.

Our proposed method, in contrast, is end-to-end in that it directly produces detections without any additional inputs besides a video clip. Moreover, it outperforms these prior works without resorting to external proposals or memory banks, showing that a transformer backbone is sufficient to capture long-range dependencies in the input video. In addition, unlike previous two-stage methods, our method directly predicts tubelets: a sequence of bounding boxes and actions for each frame of the input video, and can do so even when we do not have full tubelet annotations available.

A number of proposal-free action localisation models have also been developed (Köpüklü et al., 2019; Chen et al., 2021; Sun et al., 2018; Girdhar et al., 2019; Kalogeiton et al., 2017; Zhao et al., 2022b). These methods are based upon alternative object detection architectures such as SSD (Liu et al., 2016), CentreNet (Zhou et al., 2019), YOLO (Redmon et al., 2016), DETR (Carion et al., 2020) and Sparse-RCNN (Sun et al., 2021). However, in contrast to our approach, they have been outperformed by their proposal-based counterparts. Moreover, some of these methods (Köpüklü et al., 2019; Girdhar et al., 2019; Sun et al., 2018) also consist of separate network backbones for

Figure 2: Our model processes a fixed-length video clip, and for each frame, outputs tubelets (*i.e.* linked bounding boxes with associated action class probabilities). It consists of vision encoder which outputs a video representation, $\mathbf{x} \in \mathbb{R}^{T \times h \times w \times d}$. The video representation, along with learned queries, $\mathbf{q}$ (which are factorised into spatial $\mathbf{q}^s$ and temporal components $\mathbf{q}^t$) are decoded into tubelets by a decoder of $L$ layers followed by shallow box and class prediction heads.

learning video feature representations and proposals for a keyframe, and are thus effectively two networks trained jointly, and cannot predict tubelets either.

Among prior works that do not use external proposals, and also directly predict tubelets (Kalogeiton et al., 2017; Li et al., 2020b; Song et al., 2019; Li et al., 2018; Singh et al., 2017), our work is most similar to TubeR (Zhao et al., 2022b) given that our model is also based on DETR. Our model, however, is designed with additional temporal inductive biases which improves accuracy (without using external memory banks precomputed offline like Zhao et al. (2022b). And moreover, unlike TubeR, we also demonstrate how our model can predict tubelets (*i.e.* predictions at every frame of the input video), even when we only have sparse keyframe supervision (*i.e.* ground truth annotation for a limited number of frames) available.

Finally, we note that DETR has also been extended as a proposal-free method to addressing other localisation tasks in video such as temporal localisation (Liu et al., 2022; Zhang et al., 2021; Nawhal & Mori, 2021), instance segmentation (Wang et al., 2021) and moment retrieval (Lei et al., 2021).

## 3 SPATIO-TEMPORAL ACTION TRANSFORMER

Our proposed model ingests a sequence of video frames, and directly predicts tubelets (a sequence of bounding boxes and action labels). No external person detections (Pan et al., 2021; Wang et al., 2023; Tong et al., 2022), or memory banks (Zhao et al., 2022b; Wu et al., 2022), are needed.

As summarised in Fig. 2, our model consists of a vision encoder (Sec. 3.1), followed by a decoder which processes learned query tokens into output tubelets (Sec. 3.2). We incorporate temporal inductive biases into our decoder to improve accuracy and tubelet prediction with weaker supervision. Our model is inspired by the DETR architecture (Carion et al., 2020) for object detection in images, and is also trained with a set-based loss and Hungarian matching. We detail our loss, and how we can train with either sparse keyframe supervision or full tubelet supervision, in Sec. 3.3.

### 3.1 VISION ENCODER

The vision backbone processes an input video, $\mathbf{X} \in \mathbb{R}^{T \times H \times W \times 3}$ to produce a feature representation of the input video $\mathbf{x} \in \mathbb{R}^{t \times h \times w \times d}$. Here, $T$, $H$ and $W$ are the original temporal-, height- and width-dimensions of the input video respectively, whilst $t$, $h$ and $w$ are the spatio-temporal dimensions of their feature representation, and $d$ its latent dimension. When using a transformer backbone, these spatio-temporal dimensions depend on the patch size when tokenising the input, and when using a convolutional backbone, they depend on the overall stride. To retain spatio-temporal information, we remove the spatial- and temporal-aggregation steps at the end of the original backbone. And if the temporal patch size (or stride) is larger than 1, we bilinearly upsample the final feature map along the temporal axis to maintain the original temporal resolution.

### 3.2 TUBELET DECODER

Our decoder processes the visual features, $\mathbf{x} \in \mathbb{R}^{T \times h \times w \times c}$, along with learned queries, $\mathbf{q} \in \mathbb{R}^{T \times S \times d}$, to output tubelets, $\mathbf{y} = (\mathbf{b}, \mathbf{a})$ which are a sequence of bounding boxes, $b \in \mathbb{R}^{T \times S \times 4}$ and corresponding actions, $a \in \mathbb{R}^{T \times S \times C}$. Here, $S$ denotes the maximum number of bounding

boxes per frame (padded with "background" as necessary) and $C$ denotes the number of output classes.

The idea of decoding learned queries into output detections using the transformer decoder architecture of Vaswani et al. (2017) was used in DETR (Carion et al., 2020). In summary, the decoder of (Carion et al., 2020; Vaswani et al., 2017) consists of $L$ layers, each performing a series of self-attention operations on the queries, and cross-attention between the queries and encoder outputs.

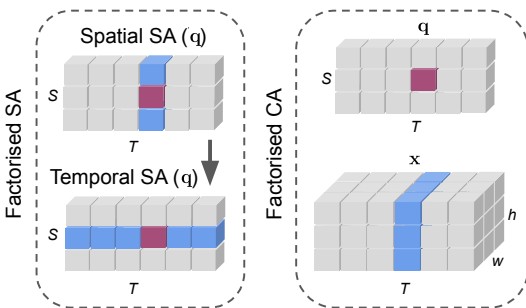

We modify the queries, self-attention and cross-attention operations for our spatio-temporal localisation scenario, as shown in Fig. 2 and 3 to include additional temporal inductive biases, and to improve accuracy as detailed below.

**Queries** Queries, $\mathbf{q}$, in DETR, are decoded using the encoded visual features, $\mathbf{x}$, into bounding box predictions, and are analogous to the "anchors" used in other detection architectures such as Faster-RCNN (Ren et al., 2015).

Figure 3: Our decoder layer consists of factorised self-attention (SA) (left) and cross-attention (CA) (right) operations designed to provide a spatio-temporal inductive bias and reduce computation. Both operations restrict attention to the same spatial and temporal slices as the query token, as illustrated by the receptive field (blue) for a given query token (magenta). Factorised SA consists of two operations, whilst Factorised CA has one.

The most straightforward way to define queries is to randomly initialise $\mathbf{q} \in \mathbb{R}^{T \times S \times d}$, where there are $S$ bounding boxes at each of the $T$ input frames in the video clip.

However, we find it is more effective to factorise the queries into separate learned spatial, $\mathbf{q}^s \in \mathbb{R}^{S \times d}$, and temporal, $\mathbf{q}^{T \times d}$ parameters. To obtain the final tubelet queries, we simply repeat the spatial queries across all frames, and add them to their corresponding temporal embedding at each location, as shown in Fig. 2. More concretely $\mathbf{q}_{ij} = \mathbf{q}_i^t + \mathbf{q}_j^s$ where $i$ and $j$ denote the temporal and spatial indices respectively.

The factorised query representation means that the same spatial embedding is used across all frames. Intuitively, this encourages the $i^{th}$ spatial query embedding, $\mathbf{q}_i^s$, to bind to the same location across different frames of the video, and since objects typically have small displacements from frame to frame, may help to associate bounding boxes within a tubelet together. We verify this intuition empirically in the experimental section.

**Decoder layer** The decoder layer in the original transformer (Vaswani et al., 2017) consists of self-attention on the queries, $\mathbf{q}$, followed by cross-attention between the queries and the outputs of the encoder, $\mathbf{x}$, and then a multilayer perceptron (MLP) layer (Hendrycks & Gimpel, 2016):

$$\mathbf{u}^\ell = \text{MHSA}(\mathbf{q}^\ell) + \mathbf{q}^\ell, \quad \mathbf{v}^\ell = \text{CA}(\mathbf{u}^\ell, \mathbf{x}) + \mathbf{u}^\ell, \quad \mathbf{z}^\ell = \text{MLP}(\mathbf{v}^\ell) + \mathbf{v}^\ell, \tag{1}$$

where $\mathbf{z}^\ell$ is the output of the $\ell^{th}$ decoder layer, $\mathbf{u}$ and $\mathbf{v}$ are intermediate variables, MHSA denotes multi-headed self-attention and CA denotes cross-attention. Note that the inputs to the MLP, self- and cross-attention operations are layer-normalised (Ba et al., 2016), which we omit here for clarity.

In our model, we factorise the self- and cross-attention layers across space and time respectively as shown in Fig. 3, to introduce a temporal locality inductive bias, and also to increase model efficiency. Concretely, when applying MHSA, we first compute the queries, keys and values, over which we attend twice: first independently at each time step with each frame, and then, independently along the time axis at each spatial location. Similarly, we modify the cross-attention operation so that only tubelet queries and backbone features from the same time index attend to each other.

**Localisation and classification heads** We obtain the final predictions of the network, $\mathbf{y} = (\mathbf{b}, \mathbf{a})$, by applying a small feed-forward network to the outputs to the decoder, $\mathbf{z}$, following DETR (Carion et al., 2020). The sequence of bounding boxes, $\mathbf{b}$, is obtained with a 3-layer MLP, and is parameterised by the box center, width and height for each frame in the tubelet. A single-layer linear projection is used to obtain class logits, $\mathbf{a}$. As we predict a fixed number of $S$ bounding boxes per frame, and $S$ is more than the maximum number of ground truth instances in the frame, we also include an additional class label, $\varnothing$, which represents the "background" class which tubelets with no action class can be assigned to.

### 3.3 TRAINING OBJECTIVE

Our model predicts bounding boxes and action classes at each frame of the input video. Many datasets, however, such as AVA (Gu et al., 2018), are only sparsely annotated at selected keyframes of the video. In order to leverage the available annotations, we compute our training loss, Eq. 2, only at the annotated frames of the video, after having matched the predictions to the ground truth:

$$\mathcal{L}(\mathbf{y}, \hat{\mathbf{y}}) = \frac{1}{|\mathcal{T}|} \sum_{t \in \mathcal{T}} \mathcal{L}_{\text{frame}}(\mathbf{y}, \hat{\mathbf{y}}), \tag{2}$$

where $\mathcal{T}$ is the set of labelled frames; $\mathbf{y}$ and $\hat{\mathbf{y}}$ denote the ground truth and predicted tubelets after matching. Following DETR (Carion et al., 2020), our training loss at each frame, $\mathcal{L}_{\text{frame}}$, is a sum of an $L_1$ regression loss on bounding boxes, the generalised IoU loss (Rezatofighi et al., 2019) on bounding boxes, and a cross-entropy loss on action labels:

$$\mathcal{L}_{\text{frame}}(\mathbf{b}^t, \hat{\mathbf{b}}^t, \mathbf{a}^t, \hat{\mathbf{a}}^t) = \sum_i \mathcal{L}_{\text{box}}(\mathbf{b}_i^t, \hat{\mathbf{b}}_i^t) + \mathcal{L}_{\text{iou}}(\mathbf{b}_i^t, \hat{\mathbf{b}}_i^t) + \mathcal{L}_{\text{class}}(\mathbf{a}_i^t, \hat{\mathbf{a}}_i^t). \tag{3}$$

**Matching** Set-based detection models such as DETR can make predictions in any order, which is why the predictions need to be matched to the ground truth before computing the training loss.

The first form of matching that we consider is to independently perform bipartite matching at each frame to align the model's predictions to the ground truth (or the $\varnothing$ background class) before computing the loss. In this case, we use the Hungarian algorithm (Kuhn, 1955) to obtain $T$ permutations of $S$ elements, $\hat{\pi}^t \in \Pi^t$, at each frame, where the permutation at the $t^{th}$ frame minimises the per-frame loss,

$$\hat{\pi}^t = \arg\min_{\pi \in \Pi^t} \mathcal{L}_{\text{frame}}(\mathbf{y}^t, \hat{\mathbf{y}}_{\pi(i)}^t). \tag{4}$$

An alternative is to perform *tubelet matching*, where all queries with the same spatial index, $\mathbf{q}^s$, must match to the same ground truth annotation across all frames of the input video. Here the permutation is obtained over $S$ elements as

$$\hat{\pi} = \arg\min_{\pi \in \Pi} \frac{1}{|\mathcal{T}|} \sum_{t \in \mathcal{T}} \mathcal{L}_{\text{frame}}(\mathbf{y}^t, \hat{\mathbf{y}}_{\pi^t(i)}^t). \tag{5}$$

Intuitively, tubelet matching provides stronger supervision when we have full tubelet annotations available. Note that regardless of the type of matching that we perform, the loss computation and the overall model architecture remains the same. Note that we do not weight terms in Eq. 3, for both matching and loss calculation, for simplicity, and to avoid having additional hyperparameters, as also done by Minderer et al. (2022).

### 3.4 DISCUSSION

As our approach is based on DETR, it does not require external proposals nor non-maximal suppression for post-processing. The idea of using DETR for action localisation has also been explored by TubeR (Zhao et al., 2022b) and WOO (Chen et al., 2021). There are, however, a number of key differences: WOO does not detect tubelets at all, but only actions at the center keyframe. We also factorise our queries in the spatial and temporal dimensions (Sec. 3.2) to provide inductive biases urging spatio-temporal association. Moreover, we predict action classes separately for each time step in the tubelet, meaning that each of our queries binds to an actor in the video. TubeR, in contrast, parameterises queries such that they are each associated with separate actions (features are average-pooled over the tubelet, and then linearly classified into a single action class). This choice also means that TubeR requires an additional "action switch" head to predict when tubelets start and end, which we do not require as different time steps in a tubelet can have different action classes in our model. Furthermore, we show experimentally (Tab. 1) that TubeR's parameterisation obtains lower accuracy. We also consider two types of matching in the loss computation (Sec. 3.3) unlike TubeR, with "tubelet matching" designed for predicting more temporally consistent tubelets. And in contrast to TubeR, we experimentally show how our decoder design allows our model to accurately predict tubelets even with weak, keyframe supervision.

Finally, TubeR requires extra complexity in the form of a "short-term context module" (Zhao et al., 2022b) and the external memory bank of Wu et al. (2019) which is computed offline using a separate model to achieve strong results. As we show experimentally, we outperform TubeR without any additional modules, meaning that our model does indeed produce tubelets in an end-to-end manner.

## 4 EXPERIMENTAL EVALUATION

### 4.1 EXPERIMENTAL SET-UP

**Datasets** We evaluate on four spatio-temporal action localisation benchmarks. AVA and AVA-Kinetics contain sparse annotations at each keyframe, whereas UCF101-24 and JHMDB51-21 contain full tubelet annotations.

*AVA* (Gu et al., 2018) consists of 430, 15-minute video clips from movies. Keyframes are annotated at every second in the video, with about 210 000 labelled frames in the training set, and 57 000 in the validation set. There are 80 atomic actions labelled for every actor in the clip, of which 60 are used for evaluation (Gu et al., 2018). Following standard practice, we report the Frame Average Precision (fAP) at an IoU threshold of 0.5 using the latest v2.2 annotations (Gu et al., 2018).

*AVA-Kinetics* (Li et al., 2020a) is a superset of AVA, and adds detection annotations following the AVA protocol, to a subset of Kinetics 700 (Carreira et al., 2019) videos. Only a single keyframe in a 10-second Kinetics clip is labelled. In total, about 140 000 labelled keyframes are added to the training set, and 32 000 to the validation sets of AVA. Once again, we follow standard practice in reporting the Frame AP at an IoU threshold of 0.5.

*UCF101-24* (Soomro et al., 2012) is a subset of UCF101, and annotates 24 action classes with full spatio-temporal tubes in 3 207 untrimmed videos. Note that actions are not labelled exhaustively as in AVA, and there may be people present in the video who are not performing any labelled action. Following standard practice, we use the corrected annotations of Singh et al. (2017). We report both the Frame AP, which evaluates the predictions at each frame independently, and also the Video AP, which uses a 3D, spatio-temporal IoU to match predictions to targets. Since UCF101-24 videos are up to 900 frames long (median length of 164 frames), and our network typically processes $T = 32$ frames at a time, we link together tubelet predictions from our network into full-video-tubes using the same causal linking algorithm as (Kalogeiton et al., 2017; Li et al., 2020b) for fair comparison.

*JHMDB51-21* (Jhuang et al., 2013) also contains full tube annotations in 928 trimmed videos. However, as the videos are shorter and at most 40 frames, we can process the entire clip with our network, and do not need to perform any linking.

**Implementation details** For our vision encoder backbone, we consider both transformer-based (ViViT Factorised Encoder (Arnab et al., 2021a)), and convolutional (CSN (Tran et al., 2019)) backbones. For ViVIT, we use the "Base" and "Large" model sizes (Dosovitskiy et al., 2021), which are typically first pretrained on image datasets like ImageNet-21K (Deng et al., 2009) and then finetuned on video datasets like Kinetics (Kay et al., 2017). We also use CSN-152 pretrained on Instagram65M (Mahajan et al., 2018) and then Kinetics following Zhao et al. (2022b). Our models process $T = 32$ frames unless otherwise specified, with $S = 64$ spatial queries per frame and latent decoder dimensionality of $d = 2048$. Exhaustive implementation details and training hyperparameters are included in the supplement. We will also release all code and models upon acceptance.

### 4.2 ABLATION STUDIES

We analyse the design choices in our model by conducting experiments on both AVA (with sparse per-frame supervision) and on UCF101-24 (where we can evaluate the quality of our predicted tubelets). Unless otherwise stated, our backbone is ViViT-Base pretrained on Kinetics 400, and the frame resolution is 160 pixels (160p) on the smaller side.

**Comparison of detection architectures** Table 1 compares our model, where each query represents a person, and all of their actions (Sec. 3.2) to the approach of TubeR (Zhao et al., 2022b) (Sec. 3.4), where there is a separate query for each action being performed. We observe that our parameterisation has a substantial impact, with our method outperforming binding to actions by 3.1 points with a ViViT backbone, and 2.1 points with a CSN backbone on the AVA dataset, therefore motivating the design of our decoder. Appendix C shows that this trend is consistent on UCF101-24 and JHMDB too.

Another architectural baseline that we can compare to is that of a two-stage Fast-RCNN model using external person detections from Wu et al. (2019), as used by (Wu et al., 2022; Feichtenhofer et al., 2019; Fan et al., 2021; Arnab et al., 2022). This baseline using the same ViViT-B backbone achieved

Table 1: Comparison of detection architectures on AVA controlling for the same resolution (160p) and training settings. Binding each query to a person, rather than to an action (as done in TubeR (Zhao et al., 2022b)) yields solid improvements. We report the mean AP for both ViViT-B and CSN-152 backbones.

|  | ViViT-B | CSN-152 |
|---|---|---|
| Query binds to action | 23.6 | 25.7 |
| Ours, query binds to person | **26.7** | **27.8** |

Table 2: Comparison of independent and factorised queries on the AVA and UCF101-24 datasets. Factorised queries are particularly beneficial for predicting tubelets, as shown by the VideoAP on UCF101-24 which has full tube annotations. Both models use tubelet matching in the loss.

|  | AVA | UCF101-24 | | | |
|---|---|---|---|---|---|
| Query | fAP | fAP | vAP20 | vAP50 | vAP50:95 |
| Independent | 25.2 | 85.6 | 86.3 | 59.5 | 28.9 |
| Factorised | **26.3** | **86.5** | **87.4** | **63.4** | **29.8** |

Table 3: Comparison of independent and tubelet matching for computing the loss on AVA and UCF101-24. Tubelet matching helps for tube-level evaluation metrics like the Video AP (vAP). Note that tubelet matching is actually still possible on AVA as the annotations are at 1fps with actor identities.

|  | AVA | UCF101-24 | | | |
|---|---|---|---|---|---|
| Query | fAP | fAP | vAP20 | vAP50 | vAP50:95 |
| Per-frame matching | **26.7** | **88.2** | 85.7 | 63.5 | 29.4 |
| Tubelet matching | 26.3 | 86.5 | **87.4** | 63.4 | **29.8** |

Table 4: Our model can predict tubelets even when the ground truth annotations are sparse. We show this by subsampling training annotations from UCF101-24. Our model sees minimal performance deterioration even when using only $1/24$ or 4% of the annotated frames.

| Sampling | Labelled frames | fAP | vAP20 | vAP50 | vAP50:95 |
|---|---|---|---|---|---|
| All frames | 458 814 | 86.5 | 87.4 | 63.4 | 29.8 |
| Every 12 | 39 237 | 85.2 | 87.2 | 63.0 | 29.3 |
| Every 24 | 20 243 | 84.9 | 86.8 | 63.2 | 28.1 |
| One per video | 2 284 | 70.2 | 77.1 | 48.5 | 20.4 |

a mean AP of 25.2, which is still 1.5 points below our model, emphasising the promise of our end-to-end approach. Note that the proposals of Wu et al. (2019) obtain an AP50 of 93.9 for person detection on the AVA validation set. They were obtained by first pretraining a Faster-RCNN (Ren et al., 2015) detector on COCO keypoints, and then finetuning on the person boxes from the AVA training set, using a resolution of 1333 on the longer side. Our model is end-to-end, and does not require any external proposals generated by a separate model at all.

**Comparison to TubeR** The second row of Tab. 1 using a CSN-152 backbone corresponds to our reimplementation of TubeR. By keeping all other training hyperparameters constant, we observe that our query binding provides an improvement of 2.1 mAP points in a fair comparison. Note that we could not use the public TubeR code (Zhao et al., 2022a), as it does not reproduce the paper's results: A higher resolution 256p model achieved only 20 mAP when trained with the public code, whilst it is reported to achieve 31.1. Exhaustive details on our attempts to reproduce TubeR with the authors' public code is in Appendix B.

**Query parameterisation** Table 2 compares our independent and factorised query methods (Sec. 3.2) on AVA and UCF101-24. We observe that factorised queries consistently provide improvements on both the Frame AP and the Video AP across both datasets. As hypothesised in Sec. 3.2, we believe that this is due to the inductive bias present in this parameterisation. Note that we can only measure the Video AP on UCF101-24 as it has tubes labelled. We also show in Appendix C that these observations are consistent on the JHMDB dataset too.

**Matching for loss calculation** As described in Sec. 3.3, when matching the predictions to the ground truth for loss computation, we can either independently match the outputs at each frame to the ground truths at each frame, or, we can match entire predicted tubelets to the ground truth tubelets. Table 3 shows that tubelet matching does indeed improve the quality of the predicted tubelets, as shown by the Video AP on UCF101-24. However, this comes at the cost of the quality of per-frame predictions, *i.e.* Frame AP (fAP). This suggests that tubelet matching improves the association of bounding boxes predicted at different frames (hence higher Video AP), but may also impair the quality of the bounding boxes predicted at each frame (Frame AP). Note that it is technically possible for us to also perform tubelet matching on AVA, since AVA is annotated at 1fps with actor identities, and our model is input 32 frames at 12.5fps (therefore 2.56 seconds of temporal context) meaning that we have sparse tubelets with 2 or 3 annotated frames.

As tubelet matching helps with the overall Video AP, we use it for subsequent experiments on UCF101-24 and JHMDB51-21. For AVA, we use per-frame matching as the standard evaluation metric is the Frame AP, and annotations are sparse at 1fps.

**Weakly-supervised tubelet detection** Our model can predict tubelets even when the ground truth annotations are sparse and only labelled at certain frames (such as the AVA dataset). We quantita-

Table 5: Effect of decoder depth on performance on the AVA dataset. Performance saturates at $L = 6$ layers.

| Layers ($L$) | 0 | 1 | 3 | 6 | 9 |
|---|---|---|---|---|---|
| mAP ↑ | 23.4 | 24.6 | 26.2 | 26.5 | **26.7** |

Table 6: Effect of the type of attention used in the decoder on AVA. Factorised attention is both more accurate and efficient (almost half of the GFLOPs per decoder layer).

| Decoder attention | mAP | GFLOPs |
|---|---|---|
| Full | 26.4 | 10.5 |
| Factorised | **26.7** | **5.3** |

tively measure this ability of our model on UCF101-24 which has full tube annotations. We do so by subsampling labels from the training set, and evaluating the full tubes on the validation set.

As shown in Tab. 4, we still obtain meaningful tube predictions, with a Video AP20 of 77.1, when using only a single frame of annotation from each UCF video clip. When retaining 1 frame of supervision for every 24 labelled frames (which is roughly 1fps and corresponds to the AVA dataset's annotations), we observe minimal deterioration with respect to the fully supervised model (all Video AP metrics are within 0.7 points). Retaining 1 frame of annotation for every 12 consecutive labelled frames also performs similarly to using all frames in the video clip. These results suggest that due to the redundancy in the data (motion between frames is often limited), and the inductive bias of our model, we do not require each frame in the tube to be labelled in order to predict accurate tubelets.

**Decoder design** Tables 5 and 6 analyse the effect of the decoder depth and the type of attention in the decoder (described in Sec. 3.2). As seen in Tab. 5, detection accuracy on AVA increases with the number of decoder layers, plateauing at around 6 layers. It is possible to use no decoder layers too: In this case, instead of learning queries $\mathbf{q}$ (Sec. 3.2), we simply interpret the outputs of the vision encoder (Sec. 3.1), $\mathbf{x}$, as our queries and apply the localisation and classification heads directly upon them. Using decoder layers, however, can provide a performance increase of up to 3.3 mAP points (14% relative), emphasising their utility.

Table 6 shows that factorised attention in the decoder is more accurate than standard, "full" attention between all queries and visual features. Moreover, it is more efficient too, using almost half of the GFLOPs at each decoder layer.

**Additional analysis** We further analyse the effect of frame resolution, and pretraining in Appendix C. As expected, we find that higher resolutions, and larger scale pretraining, using CLIP (Radford et al., 2021), improves accuracy. We make use of these observations in our following state-of-the-art comparisons. The supplement also visualises our predicted tubelets.

### 4.3 COMPARISON TO STATE-OF-THE-ART

We compare our model to the state-of-the-art on datasets with both sparsely annotated keyframes (AVA and AVA-Kinetics), and full tubes (UCF101-24 and JHMDB).

**AVA and AVA-Kinetics** Table 7 compares to prior work on AVA and AVA-Kinetics. The best previous methods relied on external proposals (Wang et al., 2022; Tong et al., 2022; Arnab et al., 2022) and external memory banks (Pan et al., 2021; Wu et al., 2022) which we outperform. There are fewer end-to-end approaches, and we outperform these by an even larger margin. Note that though TubeR (Zhao et al., 2022b) is a proposal-free approach, their best results are actually obtained with the external memory of Wu et al. (2019). Consequently, we have reported the end-to-end, and external-memory versions of TubeR ("TubeR + LTC") separately in Tab. 7. Furthermore, as detailed in Appendix B, the TubeR public code also shows additional object detection pretraining on COCO that is not used by any other prior work. Observe that we outperform TubeR using the same CSN-152 backbone, and then improve further using larger transformer backbones.

We achieve greater relative improvements on AVA-Kinetics, showing that our end-to-end approach can leverage larger datasets more effectively. To our knowledge, we surpass the best previous results on AVA-Kinetics, achieving a Frame AP of 45.1. Notably, we outperform InternVideo (Wang et al., 2022) and VideoMAE-v2 (Wang et al., 2023), which are two recent video foundational models using more powerful backbones and larger, proprietary, web-scale video datasets. Note that InternVideo consists of two different encoders, one of which is also initialised from CLIP (Radford et al., 2021). And like Wang et al. (2022), we achieve our best AVA results by training a model on AVA-Kinetics, and then evaluating it only on the AVA validation set. Moreover, note that we do not perform any

Table 7: Comparison to the state-of-the-art (reported with mean Average Precision; mAP) on AVA v2.2 and AVA-Kinetics (AVA-K). Methods using external proposals are also trained on additional object detection and human pose data. Unless otherwise stated, separate models are trained for AVA and AVA-Kinetics. * denotes the model was trained on AVA-Kinetics and evaluated on AVA. "Res." denotes the frame resolution of the shorter side. Web-scale foundational models are denoted in grey.

| | Pretraining | Views | AVA | AVA-K | Res. | Backbone | End-to-end |
|---|---|---|---|---|---|---|---|
| MViT-B (Fan et al., 2021) | K400 | 1 | 27.3 | – | – | MViT | ✗ |
| Unified (Arnab et al., 2021b) | K400 | 6 | 27.7 | – | 320 | SlowFast | ✗ |
| AIA (Tang et al., 2020) | K700 | 18 | 32.3 | – | 320 | SlowFast | ✗ |
| ACAR (Pan et al., 2021) | K700 | 6 | 33.3 | 36.4 | 320 | SlowFast | ✗ |
| TubeR (Zhao et al., 2022b) with LTC | IG65M→K400, COCO | 2 | 33.6 | – | 256 | CSN-152 | ✗ |
| MeMViT (Wu et al., 2022) | K700 | – | 34.4 | – | 312 | MViT v2 | ✗ |
| Co-finetuning (Arnab et al., 2022) | IN21K→K700, MiT, SSv2 | 1 | 32.8 | 33.1 | 320 | ViViT/L | ✗ |
| | JFT,WTS→K700, MiT, SSv2 | 1 | 36.1 | 36.2 | 320 | ViViT/L | ✗ |
| VideoMAE (Tong et al., 2022) | SSL K700 → Sup. K700. | – | 39.3 | – | – | ViViT/L | ✗ |
| InternVideo* (Wang et al., 2022) | 7 different datasets | – | 41.0 | 42.5 | – | Uniformer v2 | ✗ |
| VideoMAE v2 (Wang et al., 2023) | 6 different datasets | – | 42.6 | 43.9 | – | ViViT/g | ✗ |
| Action Transformer (Li et al., 2020a) | K400 | 1 | – | 23.0 | 400 | I3D | ✓ |
| WOO (Chen et al., 2021) | K600 | 1 | 28.3 | – | 320 | SlowFast | ✓ |
| TubeR (Zhao et al., 2022b) | IG65M→K400, COCO | 1 | 31.1 | – | 256 | CSN-152 | ✓ |
| STAR/CSN-152 (ours) | IG65M→K400 | 1 | 31.4 | 35.8 | 256 | CSN-152 | ✓ |
| STAR/B (ours) | IN21K→K400 | 1 | 30.0 | 36.6 | 320 | ViViT/B | ✓ |
| | CLIP→K700 | 1 | 33.9 | 39.1 | 320 | ViViT/B | ✓ |
| STAR/L (ours) | CLIP→K700 | 1 | 39.2 | 44.5 | 320 | ViViT/L | ✓ |
| STAR/L (ours)* | CLIP→K700 | 1 | **42.5** | **45.1** | 420 | ViViT/L | ✓ |

Table 8: State-of-the-art comparison on datasets with tubelet annotations, UCF101 and JHMDB.

| | | UCF101-24 | | | | JHMDB51-21 | | | |
|---|---|---|---|---|---|---|---|---|---|
| | Pretraining | fAP | vAP20 | vAP50 | vAP50:95 | fAP | vAP20 | vAP50 | Backbone |
| ACT (Kalogeiton et al., 2017) | IN1K | 67.1 | 77.2 | 51.4 | 25.0 | 65.7 | 74.2 | 73.7 | VGG |
| MOC (Li et al., 2020b) | IN1K → COCO | 78.0 | 82.8 | 53.8 | 28.3 | 70.8 | 77.3 | 77.2 | DLA34 |
| Unified (Arnab et al., 2021b) | K600 | 79.3 | – | – | – | – | – | – | SlowFast |
| WOO (Chen et al., 2021) | K600 | – | – | – | – | 80.5 | – | – | SlowFast |
| TubeR (Zhao et al., 2022b) | IG65M→K400 | 83.2 | 83.3 | 58.4 | 28.9 | – | 87.4 | 82.3 | CSN-152 |
| TubeR with flow (Zhao et al., 2022b) | K400 | 81.3 | 85.3 | 60.2 | 29.7 | – | 81.8 | 80.7 | I3D |
| STAR/CSN-152 (ours) | IG65M→K400 | 86.7 | 87.0 | 65.4 | 30.6 | **93.5** | **96.3** | **95.4** | CSN-152 |
| STAR/B (ours) | IN21K→K400 | 87.3 | 88.2 | 68.6 | 31.7 | 86.9 | 89.5 | 88.2 | ViViT/B |
| STAR/L (ours) | CLIP→K700 | **90.3** | **89.8** | **73.4** | **35.8** | 92.1 | 93.1 | 92.6 | ViViT/L |

test-time augmentation, in contrast to previous works that ensemble results over multiple resolutions and/or left/right flips as denoted by the "Views" column.

**UCF101-24** Table 8 shows that we outperform prior work on UCF101-24, both in terms of frame-level (Frame AP), and tube-level metrics (Video AP). We achieve state-of-the-art results with a ViViT-Base backbone, and improve further by scaling up to ViViT-Large, consistent with our results on AVA (Tab. 7). Moreover, note how we substantially outperform TubeR (Zhao et al., 2022b) using the same CSN-152 backbone. To our knowledge, we outperform the best previous reported Video AP50 number by 13.2 points. Note that as UCF videos are up to 900 frames, and as our network processses $T = 32$ frames, we follow prior works and link together tubets using the same causal algorithm as (Kalogeiton et al., 2017; Li et al., 2020b; Arnab et al., 2021b) for fair comparison.

**JHMDB51-21** Table 8 also shows that we surpass the state-of-the-art on JHMDB. Once again, we outperform TubeR (Zhao et al., 2022b) with the same CSN-152 backbone. The CSN-152 backbone outperforms ViViT in this case, possibly because this is the smallest dataset and larger backbones can overfit more easily. The videos in this dataset are trimmed (meaning that labelled actions are being performed on each frame), and also shorter. Therefore, the Video AP is not as strict as it is on UCF101-24. Additionally, as the input videos are a maximum of 40 frames, we set $T = 40$ in our model so that we process the entire clip at once without needing to link tubelets.

## 5   CONCLUSION AND FUTURE WORK

We have presented STAR, an end-to-end spatio-temporal action localisation model that can output tubelets, when either sparse keyframe, or full tubelet annotation is available. Our approach achieves state-of-the-art results on four action localisation datasets for both frame-level and tubelet-level predictions (in particular, we obtain 45.1% mAP on the challenging AVA-Kinetics dataset), outper-forming complex methods that use external proposals and memory banks. Future work is to extend our method beyond fixed action classes to open vocabularies.

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

# A   QUALITATIVE EXAMPLES

Please refer to the attached supplementary video to see qualitative examples of our predicted tubelets on AVA and UCF101-24. The video uses the "H264 MP4" codec, and has been tested on "VLC Media Player" and "QuickTime".

# B   DISCREPANCIES IN OFFICIAL TUBER CODE

## B.1   ATTEMPT TO REPRODUCE TUBER TRAINING

In order to make fair comparisons to TubeR, we attempted to train a model following the instructions in the authors' public code release.

Firstly, in order to be able to run the code, we needed to fix some programming bugs as pointed out in a Github issue.

Once these fixes were applied, we were able to run the code. However, the performance on AVA after just a few epochs was poor, as shown below in Tab. 9.

Table 9: Initial attempt at running public TubeR training code on AVA. The results are poor as the public code does not load pretrained weights into the backbone by default.

| Epochs | 1 | 3 |
|--------|------|------|
| mAP | 1.43 | 1.98 |

Looking further into the configuration files provided by the authors, we noticed that, MODEL.PRETRAINED = False, meaning that weights from the backbone were not being loaded, as shown here.

By modifying the configuration file to load the pretrained weights of the backbone, the performance did improve. But after training completed, the results were far from what was expected, as shown in Tab. 10. Concretely, we obtained a final mAP of 19.9, when we expected to achieve 31.1.

We have contacted the authors regarding this issue, and have not received any response. As a result, we were not able to use the public TubeR training code for performing fair comparisons to it, and had to use our reimplementation of it in our training framework.

Finally, note that TubeR achieve their best results with a "Long-Term Context" module adapted from Wu et al. (2019). However, this part is not included in their public code release at all.

We are committed to releasing full training code on acceptance of our paper.

Table 10: Results of running the TubeR authors' public training code on AVA, with the necessary bug-fixes and corrections as described in the text. The final accuracy is still, however, much lower than expected (mAP of 19.9, compared to the expected 31.1). As a result, we could not build upon the public TubeR code-base, and use it to make fair comparisons to TubeR.

| Epoch | 2 | 4 | 6 | 8 | 10 | 12 | 14 | 16 | 18 | 20 |
|-------|-------|-------|-------|-------|-------|-------|-------|-------|-------|-------|
| mAP | 15.08 | 18.50 | 19.59 | 20.98 | 20.43 | 21.77 | 20.03 | 20.42 | 20.79 | 19.91 |
| Expected mAP | | | | | | | | | | 31.10 |

Table 11: Comparison of computational cost to representative methods on the AVA dataset. TubeR (Zhao et al., 2022b) and Co-finetuning (Arnab et al., 2022), which use external LTC and person detector modules like other two-stage approaches, do not report FLOPs and parameters of these extra modules, like other two-stage approaches. As shown in the first row, the cost of the person detector (Faster-RCNN with ResNeXt-101 FPN) used in (Wu et al., 2019; Fan et al., 2021; Arnab et al., 2021b; Wu et al., 2022; Arnab et al., 2022; Feichtenhofer et al., 2019; Wang et al., 2022; Tang et al., 2020) is significant, as it can be more than the action localisation model itself.

| Model | AVA | Total GFLOPs | Params ($10^6$) |
|---|---|---|---|
| Person detector | – | 756 | 122.2 |
| TubeR (Zhao et al., 2022b) | 31.1 | 240 | 90.1 |
| TubeR with LTC (Zhao et al., 2022b) | 33.6 | 240 + LTC | 90.1 + LTC |
| Co-finetuning (Arnab et al., 2022) | 36.1 | 4738 + person detector | 431 + detector |
| InternVideo (Wang et al., 2022) | 41.0 | – | – |
| STAR/B 4 frames | 32.3 | **168** | 126.8 |
| STAR/B 8 frames | 33.1 | 336 | 126.8 |
| STAR/B 16 frames | 34.3 | 672 | 126.8 |
| STAR/B 32 frames | 36.3 | 1345 | 126.8 |
| STAR/L 32 frames | **41.7** | 5669 | 417.1 |

### B.2 DETR PRETRAINING

The public release of TubeR initialises the decoder part of the network using DETR pretrained weights, as shown here. However, this fact was not mentioned in the paper at all.

We have contacted the authors to ask what pretraining data was used here, but they have not responded. However, based on Github issues [1] and [2], we have inferred that the DETR was in fact pretrained on COCO.

We are not aware of any other work on AVA that has used object detection pretraining, and this initialisation therefore makes TubeR not fairly comparable to other works in the literature.

Our Tubelet decoder (Section 3.2) is randomly initialised, and does not make use of additional data unlike TubeR.

### B.3 UNCLEAR IF TUBER ACTUALLY PREDICTS TUBELETS ON AVA

Reading through the public TubeR code, we noticed that the model is specialised according to the dataset (for example [1, 2, 3]). In particular, when training on AVA, the dimensionality of the queries, $\mathbf{q} \in \mathbb{R}^{S \times d}$ where $S$ is the number of spatial queries, as shown here. On other datasets like UCF101, $\mathbf{q} \in \mathbb{R}^{T \times S \times d}$ where $T$ is the temporal dimension, as shown here. This suggests that TubeR does not predict tubelets on the AVA dataset, but rather just bounding boxes at the centre keyframe. However, Figure 5 of the TubeR paper (Zhao et al., 2022b) implies that tubelets are predicted.

We have contacted the authors to clarify the aforementioned details, but they have not responded.

In contrast, we do not instantiate our model differently for each dataset. And our model predicts tubelets even if we have only sparse keyframe supervision. This was demonstrated in Table 4 of the main paper (where we performed weakly-supervised tubelet detection), and by visualisations of our results on AVA in the attached supplementary video (Sec. A).

## C ADDITIONAL EXPERIMENTS

### C.1 COMPUTATIONAL COST

Fairly comparing the computational cost of state-of-the-art models is difficult:

TubeR (Zhao et al., 2022b) achieve their best results using a "Long-term context" (LTC) module from Wu et al. (2019), which first precomputes a "Long-term feature bank" (Wu et al., 2019) from

the entire video clip. The TubeR paper (Zhao et al., 2022b) provides no details about how the "Long-term feature bank" is computed (and their public code does not include this module either). However, the original paper (Wu et al., 2019) on the AVA dataset, precomputed features over the entire 15 minute video, to supplement the 2.5 second clip that was processed by the model. Therefore, one can assume that TubeR's LTC module adds orders of magnitude additional computation.

Moreover, all of the two-stage methods in Table 9 use an additional person detector for proposals. However, they do not report the computation cost of it at all. We analysed the cost of the Faster R-CNN with ResNeXt-101-FPN (Ren et al., 2015; Lin et al., 2017a) person-detector used in (Wu et al., 2019; Fan et al., 2021; Arnab et al., 2021b; Wu et al., 2022; Arnab et al., 2022; Feichtenhofer et al., 2019; Wang et al., 2022; Tang et al., 2020), and observed that it is actually more costly than many action detection models itself, as shown in Tab. 11.

In addition, some papers, such as InternVideo (Wang et al., 2022), do not report compute metrics either.

Table 11 shows that we can trade-off speed and accuracy by varying the number of input frames. Notably, with 4 frames at 320p resolution, STAR with a ViViT-Base backbone uses the least GFLOPs and outperforms TubeR without LTC (the only variant of TubeR for which we know the total GFLOPs). Moreover, STAR/B with 16 frames uses less GFLOPs and parameters than the person detector used by existing two-stage action localisation algorithms. STAR/B with 32 frames also outperforms Arnab et al. (2022) with less than 25% of the total GFLOPs.

## C.2 Implementation details

We exhaustively list hyperparameter choices for the models used in our state-of-the-art comparisons in Tab. 12 and 13.

Note that our model hyperparameters in Tab. 12 follow the same nomenclature from ViT (Dosovitskiy et al., 2021) and ViViT (Arnab et al., 2021a) for defining "Base" and "Large" variants.

Our experiments use similar data pre-processing and augmentations as prior work (Feichtenhofer et al., 2019; Wu et al., 2022; 2019), such as horizontal flipping, colour jittering (consistently across all frames of the video) and box jittering. In addition, we used a novel keyframe "decentering" augmentation (Sec. C.6) as our model predicts tubelets, and more aggressive scale augmentation (Sec. C.7).

We train with synchronous SGD and a cosine learning rate decay schedule. As shown in Tab. 13, we typically use the same training hyperparameters across experiments. Note that for the JHMDB dataset, we use $T = 40$ frames as input to our model, as this is sufficient to cover the longest video clips in this dataset. We also do not need to perform "decentering" (Sec. C.6) for datasets with full tube annotations (UCF101-24 and JHMDB51-21). As shown in Tab. 12, we found it beneficial to use a lower learning rate for the vision encoder of our model, as it was already pretrained, in contrast to the decoder which was learned from scratch.

## C.3 Further comparison of query parameterisation to TubeR

We extend our experiments from Tab. 1 to UCF and JHMDB in Tab. 14. We observe that our method of binding queries to people, instead of actions (as done in TubeR (Zhao et al., 2022b)), still improves here, albeit by a smaller margin. Although these datasets have one action per tube, we also need to predict when an action starts and ends within a tubelet. TubeR's approach requires an additional "action switch" (Zhao et al., 2022b), which we do not (Sec. 3.4), and so our design may aid model training.

Note that experiments were performed using the same settings as Tab. 1, namely using a ViViT-Base backbone and a frame resolution of 160p.

## C.4 Effect of resolution and pretraining

Scaling up the image resolution is critical to achieving high performance for object detection in images (Huang et al., 2017; Singh & Davis, 2018). However, we are not aware of previous works studying this for video action localisation. Table 15 shows that we do indeed observe substantial

Table 12: Model architecture hyperparameters. We used the same decoder even when scaling up the vision encoder.

| Hyperparameter | Model size | |
| --- | --- | --- |
| | Base | Large |
| *Decoder* | | |
| Number of layers | 6 | |
| Learning rate | $10^{-4}$ | |
| Hidden size | 256 | |
| MLP dimension | 2048 | |
| Dropout rate | 0.1 | |
| Box head num. layers | 3 | |
| *Encoder* | | |
| Learning rate | $5 \times 10^{-6}$ | $2.5 \times 10^{-6}$ |
| Learning rate (CLIP init.) | $1.25 \times 10^{-6}$ | |
| Patch size | $16 \times 16 \times 2$ | |
| Spatial num. layers | 12 | 24 |
| Temporal num. layers | 4 | 8 |
| Attention heads | 12 | 16 |
| Hidden size | 768 | 1024 |
| MLP dimension | 3072 | 4096 |

Table 13: Model training hyperparameters for the four datasets considered in our paper. We train with synchronous SGD and a cosine learning rate decay schedule.

| Hyperparameter | Dataset | | | |
| --- | --- | --- | --- | --- |
| | AVA | AVA-K | UCF101-24 | JHMDB51-21 |
| Epochs (training steps) | 30 (148 050) | 30 (246 690) | 30 (88 230) | 40 (6 800) |
| Batch size | | 128 | | |
| Optimiser | | Adam (Kingma & Ba, 2015) | | |
| Adam $\beta_1$ | | 0.9 | | |
| Adam $\beta_2$ | | 0.999 | | |
| Gradient clipping $\ell_2$ norm | | 1.0 | | |
| Focal loss $\alpha$ | | 0.3 | | |
| Focal loss $\gamma$ | | 2.0 | | |
| Number of spatial queries ($S$) | | 64 | | |
| Number of frames ($T$) | 32 | 32 | 32 | 40 |
| Center deviation, $\rho$: per-frame matching | 4 | 4 | 0 | 0 |
| Center deviation, $\rho$: tubelet matching | 16 | 16 | 0 | 0 |
| Stochastic depth (Huang et al., 2016) | 0.2 | 0.2 | 0.5 | 0.5 |

improvements from higher resolution, improving by up to 4.6 points on AVA. As expected, higher resolutions help more for detection at small sizes, where we follow the COCO (Lin et al., 2014) convention of object sizes. Note that AVA videos have a median aspect ratio of 16:10, and we pad the larger side for videos with different aspect ratios.

Similarly, Tab. 16 shows the effect of different pretraining datasets. Video vision transformers are typically pretrained on an image dataset (like ImageNet-21K (Deng et al., 2009)) , before being finetuned on a video dataset, such as Kinetics (Kay et al., 2017). We find that the initial image checkpoint plays an important rule, with CLIP (Radford et al., 2021) pretraining significantly outperforming supervised pretraining on ImageNet-21K (Dosovitskiy et al., 2021; Steiner et al., 2022). This improvement grows further when using a "Large" backbone.

Table 14: Further comparison of query parameterisation, using the ViViT-Base backbone at 160p resolution. Binding queries to actions is done in TubeR (Zhao et al., 2022b). We report the Frame AP in all cases.

|  | AVA | UCF101-24 | JHMDB51-21 |
|---|---|---|---|
| Query binds to action | 23.6 | 87.4 | 84.1 |
| Query binds to person | **26.7** | **88.3** | **84.7** |

Table 15: Increasing the image resolution on the AVA dataset leads to consistent accuracy improvements, primarily on small objects. APs, APm and APl denote the AP at 0.5 IoU threshold on small, medium and large boxes respectively following the COCO protocol (Lin et al., 2014). AVA videos have a median aspect ratio of 16:10, and we pad the larger side when the aspect ratio is different.

| Resolution | mAP | APs | APm | APl |
|---|---|---|---|---|
| $140 \times 224$ | 25.4 | 7.2 | 11.2 | 27.8 |
| $160 \times 256$ | 26.7 | 11.5 | 12.5 | 28.7 |
| $220 \times 352$ | 28.8 | 12.0 | 15.1 | 30.7 |
| $260 \times 416$ | 29.4 | 13.3 | 15.8 | 31.0 |
| $320 \times 512$ | 30.0 | 17.5 | 16.0 | 32.0 |

### C.5 ADDITIONAL ANALYSIS OF QUERY PARAMETERISATION AND MATCHING

Tables 2 and 3 in the paper analysed the effect of our query parameterisation, and matching in the loss calculation, on the AVA and UCF101-24 datasets. In Tab. 17 and 18, we perform these experiments on JHMDB too, and find that all our findings are consistent here as well.

### C.6 DECENTERING

The majority of prior work on keyframe-based action localisation datasets (*e.g.* AVA and AVA-Kinetics) predict only at the centre frame of the video clip, as only sparse supervision at this central keyframe is available. As our model predicts *tubelets*, we intuitively would like to supervise it for other frames in the input clip as well.

To this end, we introduce another data augmentation strategy, named "decentering", where we sample video clips during training such that the keyframe with supervision is no longer at the central frame, but may deviate randomly from the central position. We parameterise this by an integer, $\rho$, which defines the maximum possible deviation, and randomly sample a displacement $\in [-\rho, \rho]$ during training.

We found that this data augmentation strategy results in qualitative improvements in the predicted tubelets (as shown in the supplementary video). However, as shown in Tab. 19, it has minimal effect on the Frame AP which only measures performance on the annotated, central frame of AVA video clips.

Note that for datasets with full tube annotations, *i.e.* UCF101-24 and JHMDB51-21, there is no need to apply decentering, as each frame of the video clip is already annotated. We do, however, use decentering with the $\rho = 8$, when training with weak supervision on UCF101-24 (Tab. 4 of the main paper).

### C.7 SCALE AUGMENTATION

Consistent with object detection in images (Ghiasi et al., 2021; Lin et al., 2017b; Cubuk et al., 2019; Singh et al., 2018), we found it necessary to perform spatial scale augmentation during training to achieve competitive action localisation performance. As shown in Tab. 20, we found that performing "zoom out" as well as "zoom in" scale augmentation during training significantly boosts action localisation performance. This departs from the choice of performing "zoom in" only scale augmentation in previous work (Feichtenhofer et al., 2019; Wu et al., 2022; 2019).

Table 16: Comparison of pretraining for our models with ViViT-B and ViViT-L backbones on AVA using a resolution of $160 \times 256$. Larger models benefit more from additional initial pretraining.

| Pretrain | STAR/B | STAR/L |
|---|---|---|
| IN21K (Deng et al., 2009) → K400 (Kay et al., 2017) | 26.7 | 27.0 |
| IN21K (Deng et al., 2009) → K700 (Carreira et al., 2019) | 27.3 | 27.6 |
| CLIP (Radford et al., 2021) → K700 (Carreira et al., 2019) | 30.3 | **36.2** |

Table 17: Comparison of independent and factorised queries the JHMDB51-21 dataset. Factorised queries are particularly beneficial for predicting tubelets, as shown by the largest improvement in the Video AP50. Both models use tubelet matching in the loss.

| Query | fAP | vAP20 | vAP50 |
|---|---|---|---|
| Independent | 85.0 | 88.5 | 85.2 |
| Factorised | **86.9** | **89.5** | **88.2** |

## C.8 FOCAL AND AUXILIARY LOSS

Following (Minderer et al., 2022; Zhu et al., 2021; Zhang et al., 2023) we use sigmoid focal cross-entropy loss (Lin et al., 2017b) as our classification loss,

$$
\begin{aligned}
\mathcal{L}_{\text{class}}(a, \hat{a}) = & -\alpha \cdot a \cdot \hat{a}^{\gamma} \log(\hat{a}) \\
& - (1 - \alpha)(1 - a)(1 - \hat{a})^{\gamma} \log(1 - \hat{a}),
\end{aligned}
\tag{6}
$$

where $a$ and $\hat{a}$ are the ground truth and predicted action class probabilities respectively. $\alpha$ and $\gamma$ are hyperparameters of the focal loss (Lin et al., 2017b). Furthermore, following Minderer et al. (2022) we do not use auxiliary losses (Carion et al., 2020) (*i.e.* attaching output heads after each decoder layer and summing up the losses from each layer) previously found to be beneficial for matching-based detection models. Both of these choices are motivated by our ablations in Tab. 21: We observe that the focal loss consistently improves performance, and that auxilliary losses are only beneficial when the focal loss is not used.

Table 18: Comparison of independent and tubelet matching for computing the loss on JHMD51-21. Tubelet matching is particularly beneficial for the Video AP50, showing that it helps to predict more temporally coherent tubelets, as is the case on UCF101-24 as well (in the main paper).

| Query | fAP | vAP20 | vAP50 |
|---|---|---|---|
| Per-frame matching | 86.0 | 88.3 | 86.0 |
| Tubelet matching | **86.9** | **89.5** | **88.2** |

Table 19: Effect of keyframe decentering studied on the AVA dataset (resolution 160p) for a model with IN21K→K400 initialisation, and factorised queries. Mild amounts of keyframe decentering do not hurt performance measured on the center frame while explicitly supervising the models ability to localise and predict actions on other frames. In fact, models trained with small amounts of decentering tend to perform better than models trained without any decentering.

| Center deviation, $\rho$ | mAP |
|---|---|
| 0 | 26.5 |
| 1 | 26.8 |
| 2 | 26.5 |
| 4 | 26.7 |
| 8 | 26.4 |
| 16 | 26.6 |

Table 20: Comparison of spatial scale augmentation for our models with a ViViT/B backbone on the AVA dataset (resolution of 140p on the shorter side). We find that large range in scale-jittering, in the range of $(0.5, 2.0)$ of the original input frame, as used in (Ghiasi et al., 2021) works the best. Notably, doing scale augmentation in range of $(\frac{8}{7}, \frac{10}{7})$ as in the open-sourced SlowFast (Feichtenhofer et al., 2019) performs significantly worse. Performing no scale augmentation (first row) performs the worst as expected.

| Scale (min, max) | mAP |
|---|---|
| $(1, 1)$ (none) | 22.5 |
| $(1.14, 1.43)$ (Feichtenhofer et al., 2019) | 23.9 |
| $(0.25, 1.0)$ | 22.7 |
| $(0.5, 1.0)$ | 23.4 |
| $(0.25, 4.0)$ | 25.1 |
| $(0.5, 2.0)$ | **25.6** |

Table 21: Effect of using sigmoid loss and auxiliary losses studied on the AVA dataset (resolution 160p) for a model with IN21K→K400 initialisation. Focal loss ($\alpha = 0.3$ and $\gamma = 2$) clearly performs better than the alternatives. Moreover, the use of auxiliary losses leads to a mild degradation in performance when combined with focal loss, but improves results when the focal loss is not used.

| Focal loss | Auxiliary losses | mAP |
|---|---|---|
| ✗ | ✗ | 20.8 |
| ✗ | ✓ | 21.8 |
| ✓ | ✓ | 26.4 |
| ✓ | ✗ | **26.8** |

