# OpenReview forum: "End-to-End Spatio-Temporal Action Localisation with Video Transformers"
_ICLR.cc/2024/Conference — ICLR 2024 Conference Withdrawn Submission_

### Official Review · Reviewer_PJAV · 2023-10-25

**Soundness:** 3 good
**Presentation:** 3 good
**Contribution:** 2 fair
**Rating:** 5
**Confidence:** 3

**Summary:**

The authors propose a fully end-to-end, DETR-based action localization model.  Their method is a one-stage, proposal-free method. The authors factorize the queries into spatial queries and temporal queries, which allows a consistent parameterization across different datasets.

**Strengths:**

The paper is generally well-written.

The experiments look solid and many details have been provided in the paper and in the supplementary. The authors have also promised to release the code.

**Weaknesses:**

Firstly, I cannot help but feel like the proposed designs and improvements are rather small over previous works. For instance, most of the novelty comes from the spatial and temporal factorized queries, which, while practical and beneficial for experiments, is not very interesting.



For the experiments, it feels like the largest improvements came from ViViT/L backbone, on AVA and AVA-K. Similarly, the pretraining using CLIP also seems to contribute most of the improvements in UCF-101-24, and much of the improvement in AVA and AVA-K. Comparatively, the improvements (especially on AVA, AVA-K) are not significant when the pretraining settings are the same as previous works (TubeR).

But, I understand that the authors have issues reproducing their TubeR’s code, i.e., the actual improvement is currently hard to quantify. Due to the similarities between the two pipelines, I suggest the authors to run some experiments according to TubeR’s method (i.e., their action-based parameterization and no query factorization) using the exact pre-training settings in this paper, and report them, which I believe will add to the experimental contributions of this paper.





Overall, the idea is not very interesting to me, but the experiments look solid, and seems to provide a good baseline for future works.

**Questions:**

Some other questions are below.

1)	How exactly are the queries binded to each person in the video? I don’t see any such explicit constraints.
2)	How fast is the proposed method compared to other methods? I understand that there are some GFLOPs comparisons in the supplementary, but it is difficult to compare the methods due to the presence of other parts (such as LTC or person detector). Could we see a speed comparison instead?
3)	I am rather curious regarding the possible integration with other foundational models. Since the proposed method requires special designs (such as spatial and temporal query factorization), is it more difficult or easier to integrate with other video-based foundational models (as reported in Table 7), which tend to be able to perform many other tasks as well.

---

> ### Author Response · Authors · 2023-11-14
> **Response - part 1**
>
> We thank the reviewer for their review and appreciate their candid and to the point review. We were glad to see that they recognised the careful empirical evaluation of the proposed model, the clarity of writing and the amount of detail provided in the manuscript. However, we also recognise their hesitation in assessing the differences of our method with respect to prior work.
>
> **Specific responses**
>
> *Firstly, I cannot help but feel like the proposed designs and improvements are rather small over previous works. For instance, most of the novelty comes from the spatial and temporal factorized queries, which, while practical and beneficial for experiments, is not very interesting.*
>
> We would like to point out that the difference between prior work and our method extend beyond the proposed factorised query parameterisation (Table 2) and include query-person binding (Table 1) and tubelet matching (Table 3). Beyond that, to our knowledge, our method is the first end-to-end spatio-temporal action localisation method to have been shown to work with pure transformer backbones and thus capable of benefiting from large-scale pre-training that this architecture is often used for. As noted by the reviewer, this ability can be utilised to drive a lot of the performance improvements in action localisation.
>
> *For the experiments, it feels like the largest improvements came from ViViT/L backbone, on AVA and AVA-K. Similarly, the pretraining using CLIP also seems to contribute most of the improvements in UCF-101-24, and much of the improvement in AVA and AVA-K. Comparatively, the improvements (especially on AVA, AVA-K) are not significant when the pretraining settings are the same as previous works (TubeR).*
>
> We agree with the reviewers observation that the use of (large) transformer backbones exposed to a lot of data during pre-training can drive a lot of performance in action localisation. However, we note that the ability to use such backbones for end-to-end spatio-temporal action localisation has not been demonstrated previously.
>
> Beyond that, for this discussion we have performed additional experiments with a IN21K+K700 pre-trained backbone, and show in the Table below that despite lacking CLIP pre-training this backbone also leads to significant improvements on the AVA-K benchmark when compared to the Co-finetuning work using the same backbone but with additional training on > 860K videos from the MiT and SSv2 datasets.
>
> | Method  | mAP50  |
> |---|---|
> | Co-finetuning (IN21K + K700 + MiT + SSv2)  | 33.1 |
> | STAR/L (IN21K + K700)  | **33.7** |
>
>
> *[...] Similarly, the pretraining using CLIP also seems to contribute most of the improvements in UCF-101-24 [...]*
>
> While planning the experiments in Table 8 we sought to show the best results and thus opted for using the CLIP-pretrained backbone. We would like to emphasise however, that even without the use of CLIP pretraining (e.g. STAR/CNS-152 or STAR/B) we still significantly improve on previously reported results on UCF101-24 and JHMDB51-21. In fact, for JHMDB51-21 the largest improvements do not stem from a CLIP-pre-trained backbone but instead come from the same CSN-152 backbone as used in TubeR.
>
> *Comparatively, the improvements (especially on AVA, AVA-K) are not significant when the pretraining settings are the same as previous works (TubeR).*
>
> When considering the same pre-training backbone (CSN-152) as used in TubeR, the relative improvement of our method (vAP50) is 20.2% and 15.8% for UCF and JHMDB respectively. This is larger than the relative improvements (change) observed when replacing the CSN-152 backbone with the CLIP-pretrained ViViT/L backbone: 12.2% and -2.3% for UCF and JHMDB respectively. We thus believe that for these benchmarks the comparative improvements do not merely stem from using a backbone with more (CLIP) pre-training.
>
> When considering the AVA and AVA-K benchmarks, the improvement of our method over TubeR reported numbers is admittedly modest. However, as discussed in the paper (especially Supp. Section B) and the general response, TubeR numbers are not currently reproducible despite our best efforts. Nevertheless, to present TubeR as favourably as possible, in Table 7 (and 8) we reprint the numbers reported in the original publication. However, if we compare the two methods in a more controlled manner (same pre-training, but non-factorised queries, tubelet-action binding, no tubelet matching and use of “short-term context” from TubeR - i.e. our reproduction of TubeR; see Table 1), we see a significant improvement of our method.

---

> ### Author Response · Authors · 2023-11-14
> **Response - part 2**
>
> **Specific responses**
>
> *But, I understand that the authors have issues reproducing their TubeR’s code, i.e., the actual improvement is currently hard to quantify. Due to the similarities between the two pipelines, I suggest the authors to run some experiments according to TubeR’s method (i.e., their action-based parameterization and no query factorization) using the exact pre-training settings in this paper, and report them, which I believe will add to the experimental contributions of this paper.*
>
> We appreciate the reviewers' understanding of the difficulties associated with reproducing TubeR’s results - both in their code, as well as in our implementation. We find the reviewer’s suggestion to compare the two methods within our pipeline excellent, and note that this is what we already did in Table 1, which shows that our method improves over our reproduction on TubeR (which outperforms the publically available official TubeR code).
>
> However, the results presented in Table 1 were meant to serve as an ablation, and as such were performed at smaller resolution and with smaller models. We have now extended these results to include TubeR models  (our implementation) trained with the CSN-152 backbone at resolution 256. As can be seen from the Table below, our method significantly outperforms our best-effort reimplementation of TubeR.
>
> | Method  | AVA, mAP50  | AVA-K, mAP50  |
> |---|---|---|
> | TubeR (Zhao et al.)       | 31.1  | - |
> | TubeR (ours)                | 29.5 | 33.6  |
> | STAR/CSN-152 (ours)  | **31.4**  | **35.8** |
>
> *How exactly are the queries binded to each person in the video? I don’t see any such explicit constraints.*
>
> This happens through several mechanisms.
>
> First (Section 3.2), in our loss we require that each query is used to predict all action classes for a given person box rather than just a single one (i.e. multi-hot vs one-hot targets). This binds a query to all actions of a given person in each frame and empirically we observe that the same query tends to predict the same person in other frames.
>
> Second, when using tubelet matching (Section 3.3) we explicitly demand that the matching is consistent across all frames in that if a query was matched to a given person in one frame, and that person is present in other frames of the video clip, then the same query is matched to that person in all other frames. This is done during bi-partite matching at the loss computation stage.
>
> *How fast is the proposed method compared to other methods? I understand that there are some GFLOPs comparisons in the supplementary, but it is difficult to compare the methods due to the presence of other parts (such as LTC or person detector). Could we see a speed comparison instead?*
>
> We thank the author for their interest in the speed of our proposed method, and appreciate that it is difficult to assess from the GFLOPs alone, especially in the presence of external components such as person detectors or LTC. However, we note that assessing the runtime of these individual components across different codebases, implementations and hardware is a daunting task that is challenging to carry out. Moreover, because some of the methods (Co-finetuning and InternVideo) did not make their code public, and because for some components (LTC from TubeR) there is no open source implementation (official or otherwise), we simply cannot produce time estimates for them.
>
> As a pragmatic middleground we have opted for estimating the runtime of our implementation of TubeR and our method in our codebase using the same hardware (A100 GPU). We have also measured that it takes approximately 0.2 seconds to run the person detector *per frame*, which allows us to derive a very loose lower bound on the runtime of methods that require an external person detector (as the number of input frames times 0.2 sec, completely ignoring the actual runtime of the method or that they do not output tubelets). Below we reproduce (a part of) Table 11 from the Appendix, to which we now add an additional “Time” column. All times were measured for a single video clip with length given in the “Frames” column.
>
> | Model | AVA | Frames | Time, sec |
> |---|---|---|---|
> TubeR, CSN-152 (our implementation) | 29.5 | 32 |  0.4   |
> Co-finetuing (Arnab *et al.*)                    | 36.1 | 32 | > 6.4 |
> InternVideo (Wang *et al.*)                       | 41.0 | 32 | > 6.4 |
> |---|---|---|---|
> STAR/B               | 32.3 | 4    |  0.35  |
> STAR/B                                                  | 33.1 | 8    |  0.25  |
> STAR/B                                                  | 34.3 | 16  |  0.47  |
> STAR/B                                                   | 36.3 | 32 |  0.9    |
> STAR/L                                                  | 41.7 | 32  |  0.95  |
>
> As can be seen from the timings above, our model achieves a highly competitive performance-runtime tradeoff, and, as an end-to-end method that does not require externally provided boxes, is much faster than two-stage approaches.

---

> ### Author Response · Authors · 2023-11-14
> **Response - part 3**
>
> **Specific responses**
>
>
> *I am rather curious regarding the possible integration with other foundational models. Since the proposed method requires special designs (such as spatial and temporal query factorization), is it more difficult or easier to integrate with other video-based foundational models (as reported in Table 7), which tend to be able to perform many other tasks as well.*
>
> As shown in Tables 7 and 8, our method can be applied to various backbones (e.g. ViViT or CSN-152) and various pre-training. This is due to the fact that the method does not make assumptions about the backbone besides that it provides a representation that has spatial and temporal dimensions, and with the fact that architecturally the method basically consists only of a simple 6-layer decoder (no additional encoder required, as in, for example, TubeR) that is added directly on top of the backbone. As such, it can be applied to other foundational models, such as, for example, InternVideo or VideoMAE v2 to obtain an end-to-end spatio-temporal localisation model. We will amend Section 5 of the paper to emphasise the universality of our method.

---

### Official Review · Reviewer_J8Sj · 2023-10-29

**Soundness:** 3 good
**Presentation:** 3 good
**Contribution:** 3 good
**Rating:** 5
**Confidence:** 4

**Summary:**

This paper proposes a fully end-toend, transformer based model that directly ingests an input video, and outputs tubelets, which requires no additional pre-processing and post-processing. The extensive ablation experiments on four spatio-temporal action localisation benchmarks verify the effectiveness of the proposed method.

**Strengths:**

+ This paper is well-written and well-organized.
+ Good performance on the popular benchmarks.

**Weaknesses:**

- This paper does not seem to be the first work of fully end-to-end spatio-temporal localization, while TubeR has proposed to directly detect an action tubelet in a video by simultaneously performing action localization and recognition before. This weakens the novelty of this paper. The authors claim the differences with TubeR but the most significant difference is that the proposed method is much less complex.
- The symbols in this paper are inconsistent, e.g., b.
- The authors need to perform ablation experiments to compare the proposed method with other methods (e.g., TubeR) in terms of the number of learnable parameters and GFLOPs.

**Questions:**

See weaknesses.

---

> ### Author Response · Authors · 2023-11-14
> **Response**
>
> We thank the reviewer for their review, and the work and thought that went into it. We see that while the reviewer acknowledges our methods' strong performance across a number of recognised benchmarks and careful ablations, their main concern has to do with the novelty of the presented ideas, especially in relation to TubeR. We kindly refer the reviewer to the general response, but will also briefly inline the main points in this thread.
>
> **Specific responses**
>
> *This paper does not seem to be the first work of fully end-to-end spatio-temporal localization, while TubeR has proposed to directly detect an action tubelet in a video by simultaneously performing action localization and recognition before. This weakens the novelty of this paper. The authors claim the differences with TubeR but the most significant difference is that the proposed method is much less complex.*
>
> We agree with the reviewer that our work is not the first to propose tubelet prediction for spatio-temporal action localisation. We discuss the similarities and differences between our method and TubeR in detail in the paper. We also discuss in the paper that TubeR results are currently not reproducible, and that the TubeR publication and the publically available official code do not match (see Supp. Section B).
>
> Having said that, even when comparing against these non-reproducible results, we outperform TubeR using our simpler method (Tables 7-8). Furthermore, when comparing against our reimplementation of TubeR (which achieves better results than the official publicly available code), we see more significant improvements. This is shown in Table 1 and can be seen from additional experiments performed in response to reviewer zmts. For completeness we reprint the result of these experiments here.
>
> | Method  | AVA, mAP50  | AVA-K, mAP50  |
> |---|---|---|
> | TubeR (Zhao *et al.*)       | 31.1  | - |
> | TubeR (ours)                | 29.5 | 33.6   |
> | STAR/CSN-152 (ours)  | **31.4**  | **35.8** |
>
> Finally, as noted by the reviewer, one of the differences between TubeR and our proposed method is the fact that our method is much simpler. We thus propose a simpler method whilst actually increasing the performance. We hope that the reviewer agrees with us that proposing a simpler method with better results constitutes a worthwhile and novel scientific contribution, with practical advantages too.
>
>
> *The symbols in this paper are inconsistent, e.g., b.*
>
> We thank the reviewer for noticing the inconsistencies we missed, and kindly ask to provide more details to help us correct them.
>
>
> *The authors need to perform ablation experiments to compare the proposed method with other methods (e.g., TubeR) in terms of the number of learnable parameters and GFLOPs.*
>
> We thank the reviewer for this suggestion. We have in fact included such study in the Appendix,  Table 11. To summarise the table here, we can match or outperform other methods, including TubeR, with a comparable number of parameters and fewer GFLOPs.

---

> > ### Comment · Reviewer_zmts · 2023-11-16
> > **Happy with response from reviewers**
> >
> > Thank you for your pointwise response to each question. I am happy with the responses and they answered my concerns adequately.
> >
> > I think the problem of action detection is much harder and more complex than most action recognition or video understanding colleagues like to appreciate and reproducibility is a real issue. As the author repeatedly complains about the reproducibility problem with TubeR, I would like the authors to get their reproducible code out as soon as possible. If they can reduce the GPU requirement to 4 GPUs RTX 2090 or something like that then this paper really has the potential for widespread adaptability by the community. At the moment, most state-of-the-art results are coming because of a new and very expensive backbone which really hinders the adoption of such models by average academic labs and hinders the progress of the problem as a whole.

---

> > > ### Author Response · Authors · 2023-11-16
> > > **Committed to releasing the code**
> > >
> > > Thank you. We are committed to releasing code for our model, our reimplemented TubeR, and checkpoints upon acceptance of the paper, and before the actual conference.
> > >
> > > Our models can be trained on gaming GPUs, such as RTX3090 GPUs with 24GB of RAM. In fact, most of our experiments were conducted with 16GB accelerators.

---

### Official Review · Reviewer_zmts · 2023-11-01

**Soundness:** 3 good
**Presentation:** 3 good
**Contribution:** 3 good
**Rating:** 6
**Confidence:** 4

**Summary:**

The presented works address the problem of action tubelet prediction without the requirement of memory banks from similar work by Zhao et.al. The main contribution of the work is that it is able to perform well while removing the need for a memory bank when the same backbone is used. One can train this tubelet prediction when sparse annotations are available.

**Strengths:**

- The idea of the factorised query is a good one, it makes spatial temporal query search space quite tractable. Not sure if that is from the author is it borrowed from Zhao et al or others.
- Being able to predict tubelet even when sparse annotations are available is a plus.
- Removing the need of a memory bank is a good step forward toward generalisation

**Weaknesses:**

I think the paper is written well but the numbers are a bit overhyped. The proposed work is a good extension of Zhao et al 2023b (SOTA), however, the numbers in the table show that most of the dramatic improvement over SOTA is because of the use of a better transformer backbone and better pertaining. At the same time, it improves over SOTA slightly 31.1 to 31.4 without using memory banks but the decoder used in STAR is bigger.

Minor negative
The authors mentioned that they do not require any post-processing in the abstract but they do for the causal linking algorithm, which should be cited to Singh et al. (2017) because Kalogeiton et al., 2017 borrow from the aforementioned.

More or less I am happy with the paper, please try to answer the question below so I can participate in the discussion.

**Questions:**

Table one shows significant improvement over TubeR with the use of person-bound tubelets compared to action-bound. Then why the gap is so small in Table 4 between TubeR and STAR with CSN backbone?

---

> ### Author Response · Authors · 2023-11-14
> **Response**
>
> We thank the reviewer for their review, and for their insightful comments. We are also pleased to see that the reviewer recognised the simplicity of our method, the value of using our proposed factorised query parameterisation, and highlighted that our simplified design makes a step towards stronger generalisation. We also recognise the reviewer’s question about the empirical evaluation, and hope that we adequately address it below.
>
> **Specific responses**
>
> *The idea of the factorised query is a good one, it makes spatial temporal query search space quite tractable. Not sure if that is from the author is it borrowed from Zhao et al or others.*
>
> We thank the reviewer for praising the idea of using factorised queries, and would like to clarify that, as far as we are aware, we are the first work to apply this idea to spatio-temporal action localisation.
>
> *I think the paper is written well but the numbers are a bit overhyped. The proposed work is a good extension of Zhao et al 2023b (SOTA), however, the numbers in the table show that most of the dramatic improvement over SOTA is because of the use of a better transformer backbone and better pertaining. At the same time, it improves over SOTA slightly 31.1 to 31.4 without using memory banks but the decoder used in STAR is bigger.*
>
> As discussed in the general response, and Section 4.2 and Supp. Section B of the paper, we could not reproduce TubeR results neither using TubeR’s publicly released code, nor using our reimplementation of the TubeR model. Despite this, to present TubeR as favourably as possible, we show the originally published TubeR numbers in Table 7. Compared to these non-reproducible numbers that required more pretraining (TubeR requires a pre-trained DETR decoder; see Supp. Section B), the improvement of our method is relatively small. We emphasise that our method is still simpler (for example, it does not require decoder pre-training or the action switch loss).
>
> But more importantly, as shown in the following table, when comparing our method to our reimplementation of TubeR (which obtains better numbers than the publicly available official code; see Supp. Section B), we see a much more significant difference between the two methods. This result is consistent with the ablation presented in Table 1 and is an extension of the experiments from Table 1, but performed at higher resolution (256px).
>
> | Method  | AVA, mAP50  | AVA-K, mAP50  |
> |---|---|---|
> | TubeR (Zhao *et al.*)       | 31.1  | - |
> | TubeR (ours)                | 29.5 | 33.6  |
> | STAR/CSN-152 (ours)  | **31.4**  | **35.8** |
>
> *[...] but the decoder used in STAR is bigger.*
>
> TubeR uses 6 encoder and 6 decoder layers, where each encoder layer performs self-attention and each decoder layer performs self-attention and cross-attention. STAR does not require an encoder and only uses 6 decoder layers (see Table 5). As such, the overall non-backbone architecture of STAR is smaller than that of TubeR, and the decoders are comparable.
>
> *The authors mentioned that they do not require any post-processing in the abstract but they do for the causal linking algorithm, which should be cited to Singh et al. (2017) because Kalogeiton et al., 2017 borrow from the aforementioned.*
>
> We thank the reviewer for this sharp observation. We would like to clarify that our model outputs tubelets in an end-to-end manner (without any post-processing) for short video clips. However, we indeed rely on a causal linking algorithm for constructing tubelets for long untrimmed videos (such as present in UCF101). What the abstract was meant to convey is that we do not require any post-processing such as non-maximum suppression for the predicted tubelets output directly by the model. We will amend the abstract to make this clear.
>
> We will cite Singh et al. (2017) as a reference for the linking algorithm.
>
> *Table one shows significant improvement over TubeR with the use of person-bound tubelets compared to action-bound. Then why the gap is so small in Table 4 between TubeR and STAR with CSN backbone?*
>
> We assume that the reviewer meant to reference Table 7 (and not Table 4) and respond accordingly.
>
> This again has to do with the fact that in Table 7 we reprint the non-reproducible numbers from the TubeR publication. In Table 1 and the Table above we present results based on our reimplementation of TubeR, where we are able to control for all differences between the two models except for the intended ones (query binding, query parameterisation, short-term context). When using the same recipe as in Table 1 for the CSN-152 backbone at higher spatial resolution (above Table), we see a much more significant difference in performance.

---

### Official Review · Reviewer_dyfN · 2023-11-02

**Soundness:** 3 good
**Presentation:** 4 excellent
**Contribution:** 3 good
**Rating:** 6
**Confidence:** 4

**Summary:**

The paper proposes an architecture for Spatio-Temporal Action Detection in videos. The proposed architecture, namely STAR, can be trained end-to-end without the need of additional human detectors or external memory banks. The technical design is simple yet effective. Experiments are done on 4 different datasets: AVA, AVA-Kinetics, UCF-24, JHMDB. Ablation studies are thorough and enough to understand the design choice. STAR outperforms or on par with state-of-the-art methods on the four evaluating benchmarks. Written presentation is clear and easy to read and follow.

**Strengths:**

- The proposed architecture is very simple and still being effective, be on par or outperform state-of-the-art approaches. On small datasets such as UCF and JHMBD, START strongly outperforms previous methods. On larger datasets such as AVA, AVA-Kinetics, STAR also gives competitive performance.
- Solid ablation experiments: The paper provides a thorough set of ablation experiments to validate most of components / design choices.
- Written presentation is with high clarity which helps the readers easy to read and follow.

**Weaknesses:**

- On AVA and AVA-Kinetics, it seems the key recipe for STAR is using CLIP, without CLIP STAR achieves 30-31 on AVA and 35-36 on AVA-Kinetics which are much lower compared state-of-the-art (e.g., VideoMAE v2: 42.6 on AVA and 43.9 on AVA-Kinetics). Even with model with less-pre-training, i.e., Co-fine-tuning gets 36.1 and 36.2, respectively on AVA and AVA-K. Can we have a direct comparison with other method such as TubeR where TubeR is pre-trained with CLIP & K700? Also can the author(s) provide further discussion / insights about the role of CLIP, what make it useful for STAR that much?

**Questions:**

- In table 7, the paper flagged InternVideo and VideoMAE v2 as "web-scale pre-trained", what is the size / definition of web-scale? Why CLIP, JFT, or IG65M is not considered "web-scale"?

---

> ### Author Response · Authors · 2023-11-14
> **Response**
>
> We thank the reviewer for the review, and we are glad to see that they recognised our method's effective simplicity, extensive ablations supporting the method’s design, and clarity of presentation. We hope to resolve the remaining questions through this discussion.
>
>
> **Specific responses**
>
>
> *On AVA and AVA-Kinetics, it seems the key recipe for STAR is using CLIP, without CLIP STAR achieves 30-31 on AVA and 35-36 on AVA-Kinetics which are much lower compared state-of-the-art (e.g., VideoMAE v2: 42.6 on AVA and 43.9 on AVA-Kinetics). Even with model with less-pre-training, i.e., Co-fine-tuning gets 36.1 and 36.2, respectively on AVA and AVA-K.*
>
> A direct comparison to VideoMAE v2 or Co-finetuning would not be fair, as these methods differ in:
>  * The amount of video data they have been exposed to. For example, Video MAE v2 was pre-trained on “around 1.35M clips in our mixed dataset and this is the largest dataset ever used for video masked autoencoding”; and Co-finetuning was trained on WTS (a proprietary dataset; > 50M), Kinetics 700 (> 500K), Moments in Time (> 700K) and Something-Something V2 (> 160K) video datasets; whereas our method only made use of Kinetics 400 (> 200K) or Kinetics 700 for video pre-training.
> * Both, Co-finetuning and VideoMAE are two-stage methods, meaning that they require external bounding box proposals produced by object detectors applied independently to each frame at high resolution. In contrast, our method is end-to-end and does not require external box proposals during inference.
> * And, in the case of VideoMAE v2, the backbone used is a much larger model. VideoMAE v2 uses ViViT/g - a billion parameter model instead of ViViT/L used by our method.
>
> On top of that, neither VideoMAE v2, nor Co-finetuning are capable of producing tubelet predictions, even if external box proposals were provided for every frame.
>
> However, to facilitate the comparison with Co-finetuning, we have trained STAR models that use IN21K + Kinetics 700 pre-training on AVA Kinetics. Our results below show that we outperform Co-finetuning in this more controlled comparison despite being a faster end-to-end method that directly outputs tubelets, and despite the fact that Co-finetuning additionally trains on SSv2 and MiT datasets (> 860K additional videos not seen by our models).
>
> | Method  | mAP50  |
> |---|---|
> | Co-finetuning (IN21K + K700 + MiT + SSv2)  | 33.1 |
> | STAR/L (IN21K + K700)  | **33.7** |
>
>
> *Can we have a direct comparison with other method such as TubeR where TubeR is pre-trained with CLIP & K700?*
>
> Unfortunately a comparison to TubeR has proven to be challenging (see General response, as well as Supp. Section B). However, to address the reviewer's question, we have provided a more direct comparison to the Co-finetuning method (see previous part of this response and the table within).
>
> *Also can the author(s) provide further discussion / insights about the role of CLIP, what make it useful for STAR that much?*
>
> CLIP initialisation has been shown to lead to strong empirical results across a variety of tasks, such as for example semantic segmentation (Koppula *et al.*),  video classification (Lin *et al.*; Pan *et al.*) or object detection (Minderer *et al.*, 2022). We speculate that this has to do with the data that public CLIP models have been pre-trained on. However, as the details of  the exact data mixture used for pre-training these models is not available, we cannot be certain.
>
> *In table 7, the paper flagged InternVideo and VideoMAE v2 as "web-scale pre-trained", what is the size / definition of web-scale? Why CLIP, JFT, or IG65M is not considered "web-scale"?*
>
> We thank the reviewer for their sharp observation. Admittedly, we did not apply a stringent criterion to what would constitute a “web-scale foundational model”, and instead used it to highlight models that have been exposed to large amounts of video data during pre-training. InternVideo and VideoMAE v2 clearly fall in this category, as they have respectively been pre-trained on 12M and over 1M videos. However, IG65M (65M videos) and WTS (> 50M videos) pre-training should also be included in this category, and that is an oversight on our side. We will mark IG65M and WTS models as grey in Table 7, and also change the wording to emphasise large-scale video pre-training. Finally, we would like to highlight that both - VideoMAE v2 and InternVideo are models with over 1B parameters, whereas our largest models contain ~400m parameters (i.e. less than 40%).
>
>
> **References**
>
> Koppula *et al.*, 2022 “Where Should I Spend My FLOPS? Efficiency Evaluations of Visual Pre-training Methods”
>
> Lin *et al.*, 2022 “Frozen CLIP Models are Efficient Video Learners”
>
> Pan *et al.*, 2022 “ST-Adapter: Parameter-Efficient Image-to-Video Transfer Learning”
>
> Minderer *et al.*, 2022 “Simple Open-Vocabulary Object Detection with Vision Transformers”

---

### Author Response · Authors · 2023-11-14
**General response**

We thank the reviewers for their time and effort that went into reviewing our submission, and for their insightful comments and questions. Reviewers recognised our methods' simplicity (dyfN, zmts), extensive ablations and solid experimental results (dyfN, J8Sj, PJAV) and clear presentation (dyfN, zmts, J8Sj, PJAV). However, some reviewers (J8Sj, PJAV) also had questions about the novelty of the proposed method, especially relative to TubeR (Zhao *et.al*). We address this shared point here, and respond to specific reviewers’ questions in the corresponding threads.

**Novelty and comparison to TubeR**

As openly and extensively discussed in the submitted manuscript (Sections 2 and 4.2, and Supplementary Section B), we have taken numerous attempts to reproduce TubeR results - both in our codebase and the original TubeR codebase published on Github (see Supplementary Section B), highlighted discrepancies between the TubeR publication and the official code release (Supplementary Sections B.2 and B.3), and, importantly, reached out to to TubeR authors several times via Github and email for support and clarification - each time unsuccessfully.

It is challenging to compare our contributions to TubeR, as we cannot be sure exactly how TubeR results were obtained. We feel that we have gone above and beyond in making a fair comparison of our method and the non-reproducible TubeR publication, and hope to convince the reviewers of that.

As highlighted by most reviewers (including J8Sj), our method is **significantly simpler** than TubeR: it does not require external memory banks, action switches, decoder pre-training or architectural specifications for different datasets (see Supp. Section B.3), while being able to make use of both - convolutional (e.g. CSN-152 used by TubeR) and transformer-based backbones prevalent in modern large-scale pre-training. Furthermore, we show that our proposed **additional inductive bias provided by queries binding to actors** (Table 1 and Supp. Table 14), **factorised queries** (Table 2 and Supp. Table 17) and **tubelet matching** (Table 3 and Supp. Table 18) all **lead to improvements** across three challenging action localisation benchmarks. We hope that it is the reviewers’ view that proposing a simpler method that achieves comparable or better results constitutes a significant and novel contribution.

Despite issues with reproducibility of TubeR results, when comparing against it in Tables 7 and 8, we reprinted TubeR numbers presented in the original publication. This was done to allow for the most favourable presentation of TubeR. However, this also makes the relative improvement of our method seem smaller. A more fair comparison of the two methods can be found in Table 1, where we compare our method to our reimplementation of TubeR (which achieves better results than the official TubeR code release). This table shows that we significantly outperform TubeR.

However, to strengthen our empirical comparison to TubeR we follow PJAV’s suggestion and expand our experiments from Table 1. In the following Table we show that when controlling for pre-training and other experimental settings and using TubeR’s action-based query binding, query parameterisation and short-term context, our method significantly outperforms (our implementation of) TubeR. Please see the corresponding thread for details.


| Method  | AVA, mAP50  | AVA-K, mAP50  |
|---|---|---|
| TubeR (Zhao et.al)       | 31.1  | - |
| TubeR (ours)                | 29.5 | 33.6 |
| STAR/CSN-152 (ours)  | *31.4*  | *35.8* |